# 3D-printed self-healing hydrogels via Digital Light Processing

Matteo Caprioli[1,2], Ignazio Roppolo ●[1✉], Annalisa Chiappone[1], Liraz Larush[2], Candido Fabrizio Pirri ●[1,3] & Shlomo Magdassi[2✉]

Self-healing hydrogels may mimic the behavior of living tissues, which can autonomously repair minor damages, and therefore have a high potential for application in biomedicine. So far, such hydrogels have been processed only via extrusion-based additive manufacturing technology, limited in freedom of design and resolution. Herein, we present 3D-printed hydrogel with self-healing ability, fabricated using only commercially available materials and a commercial Digital Light Processing printer. These hydrogels are based on a semi-interpenetrated polymeric network, enabling self-repair of the printed objects. The autonomous restoration occurs rapidly, at room temperature, and without any external trigger. After rejoining, the samples can withstand deformation and recovered 72% of their initial strength after 12 hours. The proposed approach enables 3D printing of self-healing hydrogels objects with complex architecture, paving the way for future applications in diverse fields, ranging from soft robotics to energy storage.

[1] Department of Applied Science and Technology, Politecnico di Torino, Turin, Italy. [2] Casali Center for Applied Chemistry, Institute of Chemistry, The Hebrew University of Jerusalem, Edmond J. Safra Campus – Givat Ram, Jerusalem, Israel. [3] Istituto Italiano di Tecnologia, Center for Sustainable Future Technologies, Turin, Italy. ✉email: ignazio.roppolo@polito.it; magdassi@mail.huji.ac.il

In the last decade, many studies have been devoted to the successful modeling and shaping of functional three-dimensional (3D) hydrogels[1–3]. The significant progresses achieved will significantly impact the use of water-based materials in fields such as biomedicine. In particular, regenerative medicine could greatly benefit from this development. The manufacturing of artificial substitutes that preserve, replace, or reinforce human tissues would be a groundbreaking improvement in the quality of life[2]. To make these materials tangibly appealing for real-life applications, they must be able to replicate the natural functionality of living tissue. These soft materials have to mimic sophisticated natural architectures, with a combination of properties such as mechanical strength and cell viability[3]. Furthermore, the possibility of imparting self-healing (SH) ability could push the hydrogels beyond their structural role, extending their lifetime performance currently limited by irreversible failures[4].

Up to now, many strategies were followed to produce SH hydrogels using electrostatic interactions[5,6] or dynamic covalent bonds[7,8]. Nevertheless, it is still challenging to properly process these systems in complex 3D shapes unless it is performed through additive manufacturing (AM)[9]. Generally, SH is distinguished in two mechanisms: extrinsic and intrinsic[10]. The first one involves reservoirs of unreacted monomers embedded in the network, which acts as a sealant that fills the cracks formed during the damage[10,11]. Instead, intrinsic self-repair requires the presence of functional groups able to establish new bonds between the interfaces of the crack, often different from the pristine bonds in the polymeric network[12,13]. A convenient strategy to gather 3D shaping, printability and SH properties comprises designing hydrogels with an interpenetrated network (IPN)[14,15]. Those networks combine a rigid and robust frame, generally of non-reversible chemical bonds, with a much weaker network, mostly based on reversible physical bonds[16]. As a result, IPNs display new tailored features, such as improved toughness and flexibility while retaining the characteristics of both constituents simultaneously[17]. It has been shown that IPN hydrogels match extrusion-based 3D printing material requirements[18–20]. The viscosity of the ink, given by the interaction among the macromolecules, can be tuned to be suitable for extrusion, while the chemical network could rapidly fix the shape[21,22]. The physical network was also exploited to impart self-repairing characteristics, and this approach enabled the fabrication of self-healable 3D constructs with very simple shapes[22–24].

Material extrusion-based 3D printing technologies for hydrogels are the most common AM techniques applied in biofabrication and provided an incredible step forward in the development of customized replicas, mimicking natural structures with controlled geometry and characteristics[25]. Typically, extrusion printing shows warped and slightly deformed planes because of the shape of the extruded materials upon fixation, usually based on rheology or solidification of the material[26,27]. In addition, these technologies are limited by the requirements for high ink viscosities and structural deformations or failures depending on the object orientation[18,26–28]. Extrusion-based printing also commonly operates at low building speed and resolution (Supplementary Table 1)[29]. On the contrary, vat photopolymerization (VP) is capable of fabricating 3D hydrogels with higher geometrical complexity and finer accuracy, with no substantial spatial resolution effect on the printing time[26,29,30]. In VP, the projection of light triggers a localized solidification of a liquid formulation, layer-by-layer, leading to the fabrication of precise self-standing 3D structures with fine spatiotemporal control[31–33]. Among VP technologies, Digital Light Processing (DLP) is of particular interest due to its speed and resolution. DLP is a sequential layer-by-layer maskless photolithography technique, in which an entire slice of the object with controlled thickness is selectively solidified in a single exposure by a UV or visible light projection of a 2D pattern on a liquid photocurable resin. The build platform is then moved to fill the printing area with the uncured resin before fabricating the next layer, repeating the process until complete fabrication. In DLP it is possible to obtain flat vertical surfaces with negligible distortion and great shape fidelity because the conversion of the liquid into a solid is very fast, since it is based on a photopolymerization reaction. In vat polymerization, unlike the extrusion-based process, there is no need for support material, and therefore lattices with overhanging features and through-holes show clean and sharp edges[26]. This is particularly significant for 3D printing of hydrogels, for which complex architectures are difficult to obtain via extrusion-based technologies[26,30,33]. Moreover, there are fewer limitations on the viscosity range suitable for the fabrication process, but the material choices are limited to liquid or soluble photopolymers, which might be a disadvantage[18,34]. Low viscosity allows increasing the amount of solvent in the inks up to 90% in volume without compromising the final precision[35]. Unfortunately, the material requirements for SH hydrogels do not seem to match light-activated 3D printing[36,37]. Vat photopolymerization relies on reactions that yield a densely chemically cross-linked network to enable the accurate tuning of the macrostructure and mesostructure simultaneously[30,38]. At the same time, the SH mechanism depends on long-range chain mobility and reversible interactions. Its efficiency decreases proportionally with the increase of the cross-linking density because it hinders the migration of macromolecules through the surfaces[39,40].

Furthermore, hydrogels usually possess either mechanical robustness or rapid self-recovery properties. Both characteristics are typically not present together in a hydrogel, which makes challenging to construct self-standing complex and high-resolution structures combined with intrinsic SH[36,40]. Finally, a major challenge is the essential presence of water, which allows for a facilitated interdiffusion of the macromolecules. On the other side, water enlarges the interchain distances, which lowers secondary forces and limits self-repair[36,40]. Therefore, it is crucial to find the right balance between cross-linking density and water content to enable an effective SH process.

There are few reports of VP-3D-printed objects fabricated with intrinsically SH polymeric materials, related mainly to elastomers. These systems are based on the diffusion of a mending agent, such as Poly (caprolactone) upon heating and cooling[41], electrostatic interactions, such as hydrogen bonds[42] and ionic bonds[43], or dynamic covalent chemistry, such as disulfide bonds[44]. However, to the best of our knowledge, there are no reports on 3D-printed hydrogels produced via VP with self-repairing properties due to the chemical, physical, and technological limitations described above.

For these reasons, here we introduce the 3D printing of SH hydrogels obtained through VP. We successfully merged the intriguing characteristics of the VP printing process with SH properties to fabricate 3D-printed hydrogels with complex structures and SH ability at room temperature without any stimuli. These results were achieved by using only commercially available compounds which are able to effectively interact via dispersive forces such as Poly (vinyl alcohol) (PVA) and photocurable species, like acrylic acid (AAc), and Poly (ethylene glycol) diacrylate (PEGDA)[45–48], while the printing is performed using a commercial DLP printer. This work offers a general and easily adaptable approach to develop self-repairing hydrogels with complex 3D architecture via VP for applications in diverse fields, ranging from biomedicine and wearable sensors to robotics and energy harvesting.

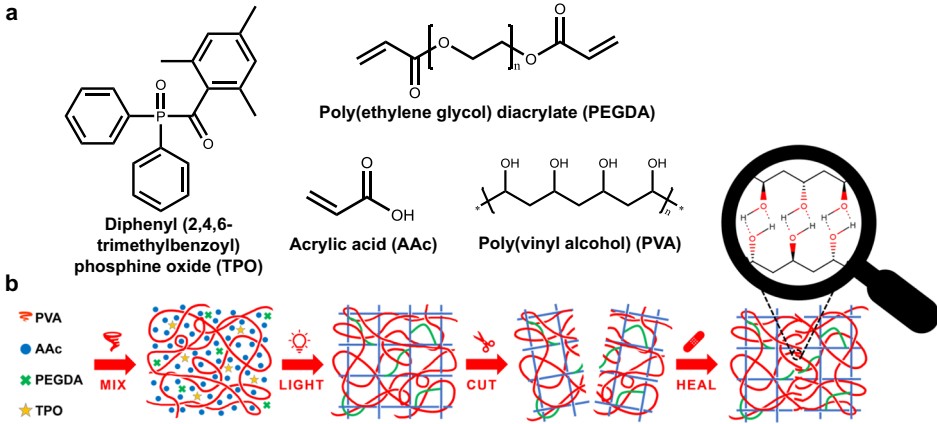

**Fig. 1 Formulation composition and network formation. a** Chemical structure of initiator, monomer, cross-linker, and mending agent in the photocurable resin. **b** Schematic illustration of the semi-interpenetrated network (semi-IPN) and healing process.

## Results

**System design**. The system was designed as a sequential semi-IPN by adding the precursors of the chemical covalent network to a solution containing a linear polymer. After polymerization, the linear polymer is entrapped in the cross-linked matrix[17]. The photocurable ink was prepared by mixing an aqueous solution of unmodified non-crosslinked Poly (vinyl alcohol) (PVA) with acrylic acid (AAc), the cross-linker Poly (ethylene glycol) diacrylate (PEGDA), and a water-compatible photoinitiator based on diphenyl (2,4,6-trimethylbenzoyl)phosphine oxide (TPO)[49] (Fig. 1.a). Organic water-soluble dyes (details in the "Methods" section) were added to improve the printing resolution, as shown in Supplementary Fig. 1. PVA was selected because of its inherent properties, such as chain interdiffusion and the presence of hydrogen bond-forming hydroxyl groups, which make PVA an attractive material for applications in efficiently SH and biocompatible hydrogels[45–48,50,51]. In preliminary experiments, the optimal molecular weight (MW) of the PVA for our application was determined to be 89-98KDa (Supplementary Fig. 2). PVA with a lower MW did not show adequate SH properties even at a high weight percentage of 30% in water, which is the highest weight percentage that still provides suitable viscosity for DLP. Likewise, higher MW did not show acceptable results because of the high viscosity even at low concentrations. The selected water-soluble acrylates, AAc and PEGDA, have been widely exploited for VP to fabricate high-resolution objects because of their ability to undergo rapid photoinduced radical photopolymerization in large amounts of water[52,53]. In addition, the carboxylic groups of the acrylic acid can form multiple hydrogen bonds with the PVA chains and, hence, contribute to the SH ability[54]. Furthermore, a considerable amount of water prevents the use of many commercial photoinitiators because most of those which match the wavelength used in VP printing show low solubility in water[55]. Therefore, the application of our developed water-compatible photoinitiator nanoparticles was crucial to achieve rapid printing using a commercial printer[49,52].

Figure 1b depicts the formation of the semi-IPN: the physical network is mixed with the precursors of the chemical network which is formed during light irradiation[17,34]. The result is a hydrogel composed of PVA chains that are homogeneously distributed and incorporated within a cross-linked acrylic matrix. The homogeneity is confirmed both by the transparency of the formulations and printed objects, and by ATR-IR spectra collected in different points of dried samples, in which typical peaks of both PVA and AAc are present simultaneously (Supplementary Fig. 3 and Supplementary Table 2). This semi-IPN should allow the system to recover from damages: once the

covalent bonds are broken, the damage can be overcome by restoring the reversible physical cross-linking amid the cut interfaces[36]. Furthermore, as it will be shown, the self-repair occurs without external stimuli or adhesives and is only due to internal forces. This is because the printed material itself is already rich in functional groups, which are able to form hydrogen bonding. Furthermore, the hydrogel contains a large proportion of water, providing PVA mobility to migrate across the ruptured interface[7,45]. At last, these secondary forces should be strong enough to be effective despite the significant amount of water in the gel[45,46].

**DLP 3D printing**. To assess the suitability of the proposed system as ink for DLP printing, we evaluated the effect of the relative concentrations of PVA and AAc, to control the extent of secondary forces and to define the printability window. The compositions of the tested formulations are presented in Supplementary Table 3. We found that the viscosity of the resulting solutions drastically increased with increasing concentration of PVA (Supplementary Fig. 4), and the solutions could not be effectively printed above a PVA/AAc ratio of 0.8 (wt/wt).

We also conducted preliminary experiments to determine the effects of the PVA concentration on the photopolymerization kinetics through photo-rheology experiments (Fig. 2.a). The polymerization process occurred in <10 s, which is well within the acceptable range for high-resolution DLP printing, without significant differences in the reaction onset or rate. This indicates that the presence of PVA does not decrease the reactivity of the acrylate species. As expected, differences in the initial G' storage shear modulus values and different final plateau values were observed for the different PVA concentrations, in agreement with the rheological measurements in Supplementary Fig. 4. Furthermore, we found that the effect of the photoinitiator concentration on the kinetics of the reaction was negligible (Fig. 2b). Since the SH performance is expected to increase with increasing PVA content, we will present here mostly the results related to the 3D printable ink containing the highest PVA concentration, i.e., PVA:AAc weight ratio of 0.8. In the printing compositions, we used only one concentration of the chemical cross-linker, PEGDA, at a molar ratio of 1:1000 to acrylic acid. This was the lowest concentration of cross-linker that enabled successful printing. Increasing concentration of PEGDA results in a higher cross-linking density, which still enabled printing 3D objects but affected negatively the SH properties. The apparent values for cross-linking, including physical and chemical bonds, were

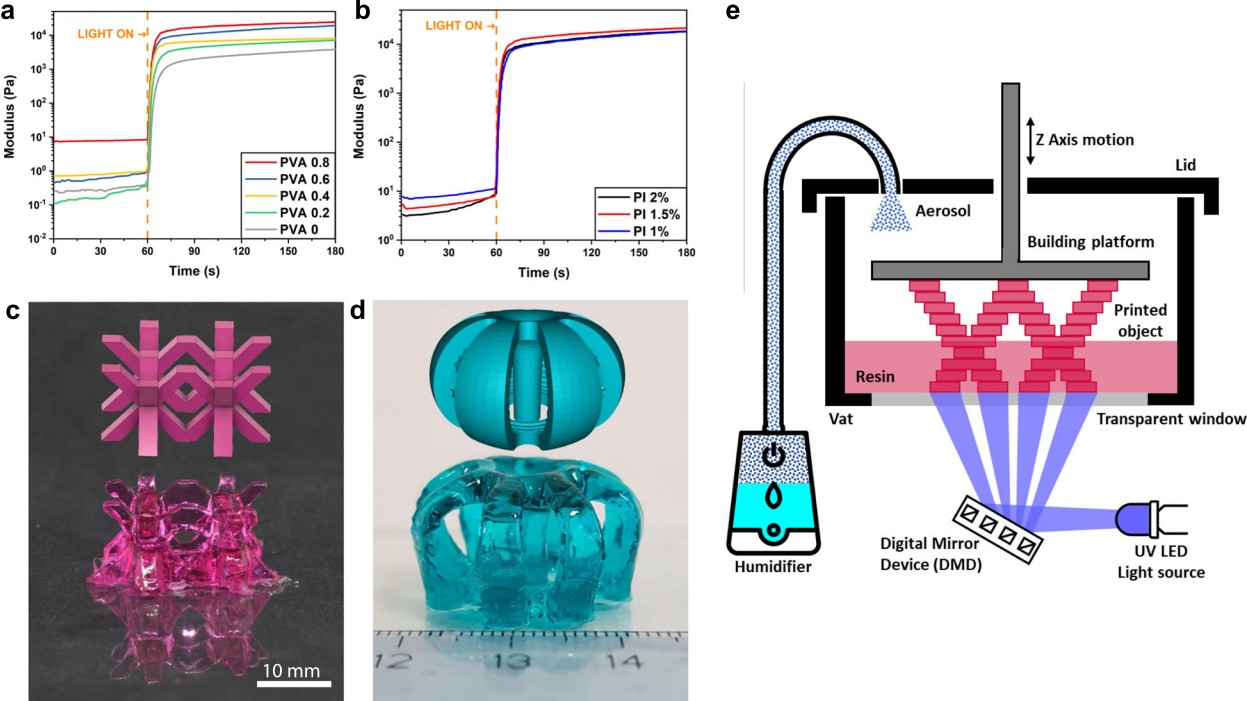

**Fig. 2 Layer-by-layer fabrication. a** Real-time photorheology of formulations with different PVA/AAc ratios under constant shear frequency (10 rad/s) and constant strain amplitude (1%). **b** Real-time photorheology of the formulation with a PVA/AAc ratio of 0.8 (PVA 0.8) with different photoinitiator concentrations under constant shear frequency (10 rad/s) and constant strain amplitude (1%). **c, d** 3D fabricated samples with PVA 0.8 formulation: **c**) Body-centered cubic lattice-like structure printed with methyl red sodium salt dye. **d** Axisymmetric structure with central pillar printed with brilliant green dye. **e** Schematic overview of the Digital Light Processing (DLP) printing apparatus used in this work.

estimated from the elastic shear modulus at the plateau of the formulations, using Eq. (1):[56]

$$\nu_e = \frac{G'_p N_A}{RT} \qquad (1)$$

where $\nu_e$ is the network cross-linking density, $G'_p$ is the shear storage modulus in the frequency-independent plateau region, $R$ is the universal gas constant, $T$ is the temperature, and $N_A$ is the Avogadro's number. We determined the crosslinking density for formulations without PVA to be $4.6 \times 10^{23}$ m$^{-3}$, and for the formulation with the highest PVA concentration, to be $4.0 \times 10^{24}$ m$^{-3}$. Supplementary Table 3 shows the calculated values for the various PVA ratios.

It should be noted that despite the very high amount of water (64%wt of the total amount) and the high viscosity, we were able to fabricate 3D-printed structures with a good CAD fidelity, as shown in Fig. 2c, d, and Supplementary Fig. 5. The printed objects are characterized by flat surfaces, straight elements, and clean and sharp edges (Fig. 2c). Objects which are difficult to print, for example, objects having overhanging features and through-holes, could be printed with this composition without any support. As an example of a complex structure, we printed objects with rotational symmetry and independent self-standing segments, with smooth rounded surfaces and a central pillar, which are shown in Fig. 2d and Supplementary Fig. 5.

It should be also noted that this high structural quality could be accomplished only after overcoming several printing-related obstacles of this formulation. The main crucial challenge was to avoid the evaporation of water during the exothermic polymerization reaction. This causes the thickening of the printable ink related to the gradual increase of PVA concentration at the interface and interferes with the layer separation/approach step. At last, the high photoreactivity of the monomer led to the

thickening of the ink and eventually uncontrolled polymerization, with loss of printing precision.

We overcame this challenge by carefully flushing a water aerosol over the printing area during printing (Fig. 2e and Supplementary Fig. 6). The optimized parameters are reported in the method section. Furthermore, the motion of the building platform should be very slow to avoid the incorporation of air bubbles in the ink, which can lead to detachment of the printed object and interlayer delamination, and to give enough time to the resin to flow and fill the gap for printing the following layer.

In addition, to avoid polymerization over the exposed areas and to achieve a better resolution, a water-soluble dye was added. This lead to better confinement of the polymerization at the x-y plane, lower light penetration depth, and better control over the photoinitiation kinetics[34,57,58]. To demonstrate this effect, cuboid samples ($25 \times 10 \times 5$ mm) were printed with formulations containing two different dyes, methyl red (left) and brilliant green (right), and then compared to a sample printed without adding a dye (Supplementary Fig. 1). The sample printed with no dye shows extensive polymerization and its shape is undefined, while the addition of a dye results in good shape fidelity with sharp edges and flat surfaces. As depicted in Supplementary Fig. 1, the dye improves the resolution in the XY plane, providing a better final quality of the printed objects[57]. At the same time, the resolution along the $Z$-axis is not only related to the dye concentration but is also strongly affected by the ink viscosity (and, therefore, by the PVA concentration) that results in a tendency to delaminate the layers immersed in a thick formulation because of an increase in the adhesion forces (Supplementary Fig. 7).

Given the considerations above, the smallest feature we achieved is around 1 mm (Supplementary Fig. 8). The obtained resolution is unfortunately well above 37 µm, the nominal maximum resolution declared for the commercial DLP printer used, mostly because of the very high PVA concentration and

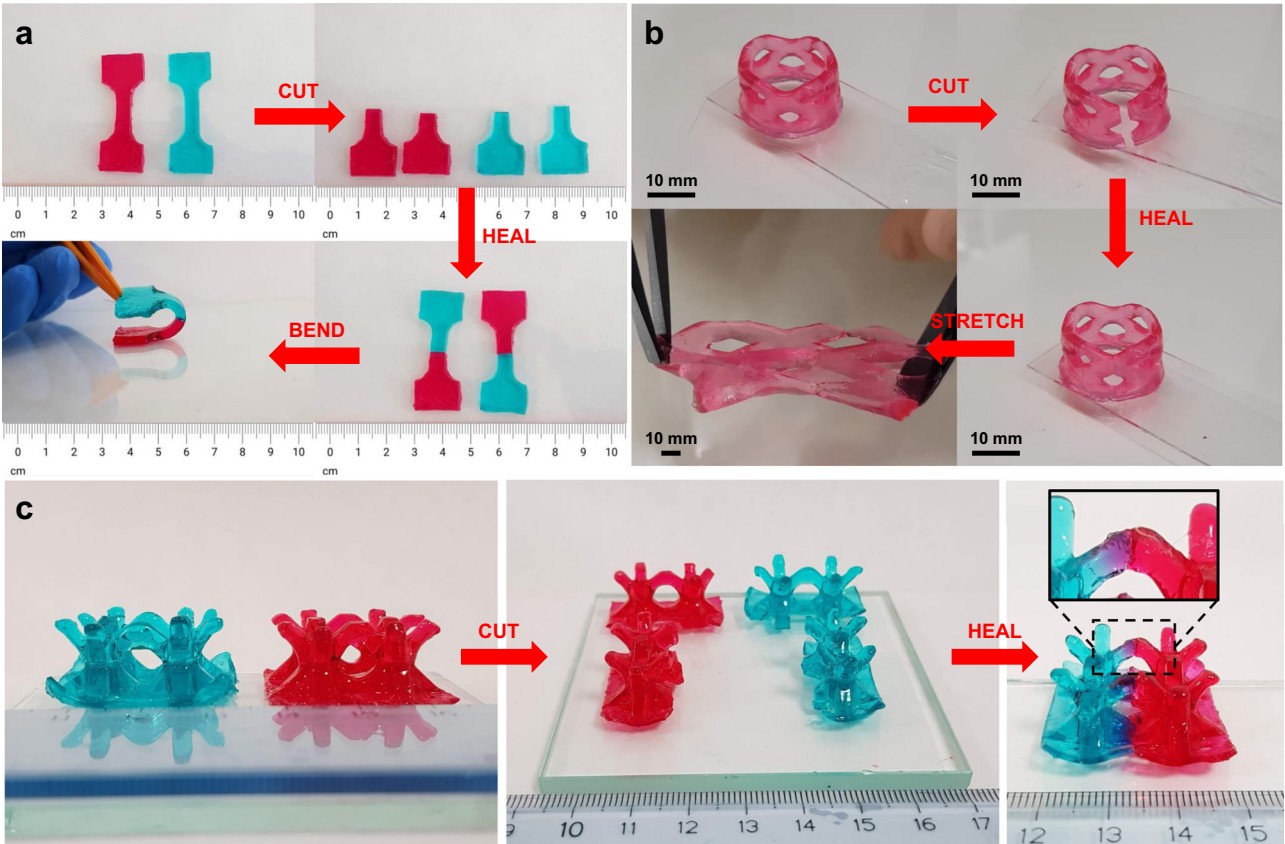

**Fig. 3 Restoration of cut and rejoined objects printed with a ratio of AAc/PVA of 0.8. a** Tensile test specimens printed with methyl red sodium salt dye and brilliant green dye. The samples could withstand a bending deformation immediately upon rejoining. **b** Holed cylindrical structure printed with methyl red sodium salt dye. The reconnected sample could endure a stretching deformation after healing for 2 h. **c** Body-centered cubic lattice-like structures printed with methyl red sodium salt dye and brilliant green dye for better visual comprehension. Diffusion at the interface after 12 h of contact is clearly visible (inset) because of the gradient of the dye.

large water amount that limit the 3D printing of small features. Although the dimensional accuracy of these printed samples is in good agreement with previous SH hydrogels 3D printed via deposition-based printing, the presented system enables better precision, CAD fidelity, shape fixity, and edge sharpness[15,20,26].

**SH of 3D-printed samples and mechanical characterization.** All SH tests were performed with the formulation containing PVA in a 0.8 weight ratio to AAc, which was the highest amount of PVA that we were able to print. Several 3D-printed objects having different shapes were cut into two pieces, and then the two parts were rapidly placed in contact to demonstrate qualitatively the SH properties. For better visualization of the SH, two samples printed with different colors, red and green, were cut and rejoined (Fig. 3). As shown, the structures were indeed macroscopically repaired. The adhesion happened instantly upon rejoining, and the mended hydrogel could immediately hold its weight and bear bending deformation without breaking apart (Fig. 3a). After 2 h, the healed sample could withstand a moderately large stretching deformation before failure (Fig. 3b and Supplementary Movie 1). The SH can be attributed to the efficient adhesion at the rejoined surface, which is facilitated by hydrogen bond formation between carboxylic and hydroxyl groups across the interface[45,54]. Furthermore, additional experiments showed that the SH on freshly cut surfaces is indeed more efficient than simple adhesion forces (Supplementary Movie 2). This behavior can be explained by a considerable amount of hydrogen bond-forming groups which

are available on the freshly cut surface and strengthen the interactions at the interface. It can be argued that the SH mechanism is similar to the behavior of pressure-sensitive adhesives (PSA) between two separate surfaces. A common measure for this effect is the Dahlquist criterion, which requires that the elastic shear modulus G′ should be <0.1 MPa[59]. Our material shows rheological properties below the Dahlquist criterion, therefore the mechanism of SH can also be influenced by surface self-adhesiveness at the interface. However, this aspect does not exclude that the material is SH, since macroscopic self-repairing can be defined as the recovery of the initial mechanical properties[36,40]. Indeed, the restoration of complex architectures with overhanging features was successfully achieved (Fig. 3c).

Tensile tests were performed at different SH times to give a quantitative temporal evaluation of the SH process (Fig. 4a). As seen, the system considerably recovers already within the first two hours. After this, the recovery slightly increases with time, reaching a plateau in the achievable restoration after 12 h of contact (Fig. 4b). The increase with time may be a result of the PVA interdiffusion[45]. The possibility of diffusion beyond the interface plan was also evaluated qualitatively using two water-soluble dyes in different sides. After rejoining, purple color is observed at the interface. (Fig. 3c, inset), which results from mixing the red and green dyes (Supplementary Fig. 9a). Note that the two dyes have different diffusivity in the hydrogel, and therefore the purple color appears to be only at one side of the interface (Supplementary Fig. 9b). It should be noted that the strong similarity between hydrogen bonding among PVA chains and the H-bonds in water in FT-IR

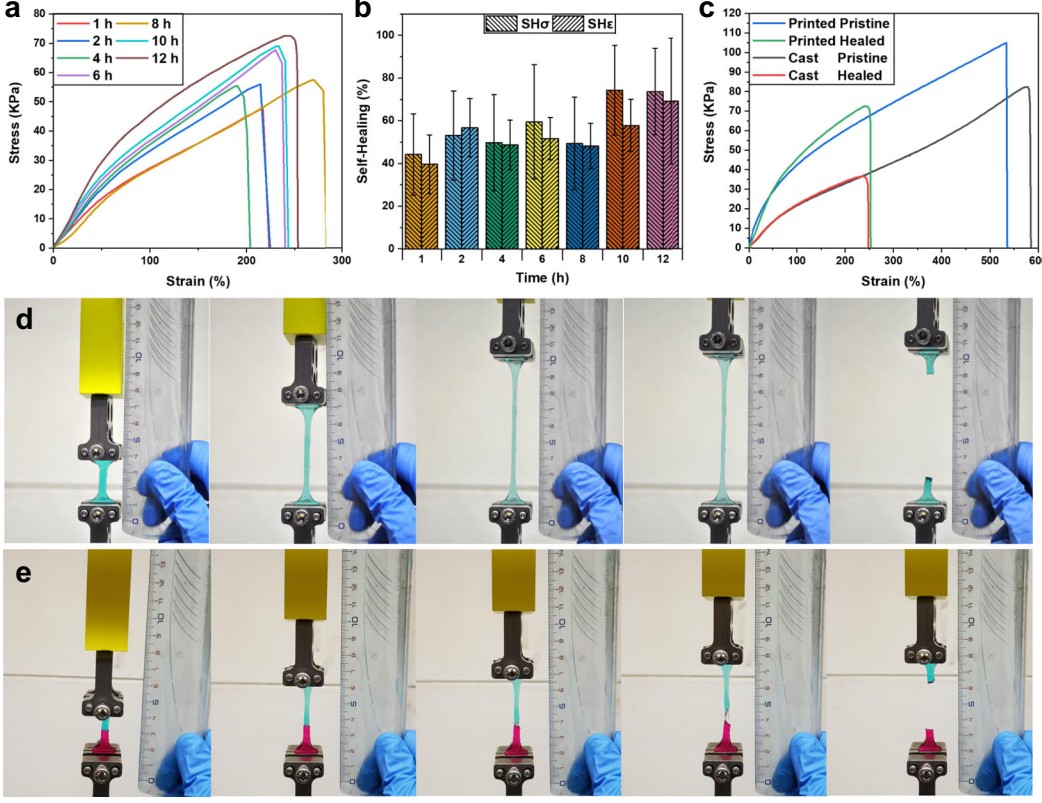

**Fig. 4 Uniaxial tensile test of specimens made with PVA 0.8 formulation. a** Stress–strain curves of the self-healed hydrogels for increasing healing time and **b** comparison of the healing efficiencies estimated from tensile strength at break (SHσ) and elongation at break (SHε). Error bars represent standard deviation, $n = 5$ independent replicates. **c** Stress–strain curves of printed pristine and healed samples compared with the same curves obtained from cast pristine, and healed samples. **d**, **e** Elongation and rupture behavior during the test of **d** pristine sample and **e** 12 h healed sample (bottom).

(broad peak around $3200\,cm^{-1}$) makes it impossible to quantitatively analyze the polymer chains diffusion or the formation of hydrogen bonding across the surfaces (Supplementary Fig. 10). In fact, the bands of hydroxyl groups are hidden by the dominant bands of water (65 wt% water in the hydrogel), as already reported in literature[45]. The variability in the stress–strain curves shown in Fig. 4b may be attributed to the irregular rupture of the healed samples that failed at the interface upon elongation, as expected (Fig. 4d). Pristine samples abruptly separated after reaching the ultimate tensile strength. In contrast, the healed samples did not split at once but gradually detached along the cut region. This feature imparts an additional safety mechanism to the repaired samples because it avoids a sudden detachment of less robust mended interfaces.

Once the restoration time was fixed, we compared the tensile strength and elongation at break of self-healed objects with pristine uncut objects. Healing efficiency was calculated from the ratio of tensile strength at break (SHσ) and elongation at break (SHε) of the rejoined objects to the values of the pristine objects. Printed dog-bone samples (0.8 weight ratio PVA/AAc) showed an average of 72% recovery in tensile strength after healing for 12 h, in good agreement with the results reported for neat PVA-based non-3D-printed hydrogels, in which the addition of a chemically cross-linked network had detrimental effects on the SH[45,46]. It is interesting to note that the SH of the printed objects was better than specimens prepared by casting in a dumbbell-shaped mold and UV-irradiated in bulk for 2 min, which regained only around 50% of the initial strength (Fig. 4c). This difference may be explained by a more homogeneous cross-linking density due to the layer-by-layer fabrication compared to photopolymerization in bulk, which is strongly affected by the light penetration depth. The samples were

kept in a sealed closed vessel in a humid environment to reduce water evaporation and maintain controlled conditions after printing during the entire healing process. This solution was crucial to preserve the SH properties, as also proven by control experiments (Supplementary Fig. 11). It should be noted that the objects possess repeatable mechanical restoration (Supplementary Fig. 11a). A second set of experiments was performed, keeping the samples in a sealed container without controlled humidity, evidencing a lower recovery efficiency every cycle (Supplementary Fig. 11b). This behavior can be explained by the loss of water during setting up the sample to measure the SH (Supplementary Fig. 11c). As expected, keeping a constant water concentration in the hydrogel is crucial to preserve the SH properties.

## Discussion

In summary, we demonstrate here 3D printing of hydrogels with SH properties via VP. This approach allows to fabricate objects with a complex architecture, including overhanging and hollow features, with sharp edges, otherwise not achievable with conventional extrusion-based printing processes. We tackled the inherent incompatibility between VP and SH properties, i.e., high cross-linking density vs. large chain mobility, using a water-based photocurable formulation that generates a physical-chemical semi-interpenetrated network. The system comprises only commercial materials and combines a covalent network of acrylic acid and Poly (ethylene glycol) diacrylate with a physical network of Poly(vinyl alcohol). Furthermore, we use a commercially available 3D DLP printer for the fabrication of the hydrogel objects. We provide a solution to challenges such as the high viscosity of the resin and the drying at the interface by performing the printing process under a water aerosol flux, to maintain a constant humidity above the vat.

SH was accomplished only at the highest printable concentration of PVA, given the viscosity limitations, indicating that there must be a trade-off to gather printability and self-repair. An insufficient amount of PVA in the hydrogel prevents the possibility of holding its weight after being cut and brought into contact, so the objects could not withstand the tensile test and failed immediately regardless of the healing time. Furthermore, if the PVA amount is too low, the chains result surrounded by an excessive volume of water molecules, weakening the adhesion forces between the two parts in contact. At this stage, to describe in details the SH, the H-bonding mechanism is a hypothesis that needs to be investigated in more detail in the future, overcoming the problem of distinguishing such bonds in systems containing a very high water concentration. Objects printed with the formulation containing the highest printable PVA amount showed rapid SH at room temperature without the need for external stimuli. Immediately after joining two cut halves, they were instantly able to withstand bending deformation without tensile stress applied, and after 12 h of healing, the samples recovered 72% of their initial tensile strength. The presence of PVA introduces an additional safety mechanism to the repaired samples because abrupt ruptures are avoided. Overall, in this study, we demonstrated that 3D-shaped SH hydrogels with very complex architectures can be fabricated through precise control over the printing composition by a low-cost commercial 3D printer and commercially available materials. The complexity related to stringent reaction conditions and complex synthesis and purification procedures for SH materials hinders mass production and economic practicability and, together with the difficulties related to the shaping, the potential applications are restricted[60]. The processing of commercially available materials via a widely available apparatus to fabricate 3D-printed hydrogels would enable fast adaptation of the proposed approach to open the way for new applications in a variety of fields, ranging from biology[2,16,34], and implantable sensors[61] to soft robotics[62,63] and energy storage[64].

## Methods

**Materials**. Poly (vinyl alcohol) (PVA) (MW 89,000-98,000 Da, 99 + % hydrolyzed); Acrylic Acid (AAc), Water-soluble TPO based nanoparticle photoinitiator (containing ionic surfactant), methyl red sodium salt (MR), and brilliant green (BG) were purchased from Sigma Aldrich (USA). Poly (ethylene glycol diacrylate) (PEGDA–SR610) was purchased from Sartomer-Arkema (France). All chemicals were used as received without further purification. The deionized (DI) water used here was purified with a Milli-Q system (Millipore, USA).

**3D ink preparation and 3D printing**. In a typical procedure, various amounts of PVA powder (Supplementary Table 2) were added in deionized water (16 mL) placed in a container immersed in iced water (4 °C) under magnetic stirring at 150 rpm for 15 min. The temperature was then gradually raised to room temperature (RT), keeping constant stirring at 150 rpm for 30 min to avoid insoluble aggregates. Afterward, the container was transferred to an oil bath at ~90 °C, maintaining the magnetic stirring at 100 rpm for 1 h. At the end of this process, the PVA was successfully dissolved in water, and the solution appeared homogeneous. Then the solution was slowly cooled down at RT. Afterward, SR 610 (50 mg), AAc (5 g), dye (1.5 mg), and photoinitiator (75 mg) were added to the solution at RT, and the whole formulation was thoroughly mixed with a centrifugal mixer (THINKY Mixer AR-100 and AR-250) for 5 min at 400 × g (2000 rpm). For naming the samples, the relative ratio between PVA and AAc in the different samples (e.g., PVA%wt/AAc%wt = 0.8 is sample PVA_0.8) was used.

Pre-designed STL models were 3D printed using a DLP printer (Asiga Pico 2HD). The DLP printing system operates with a 385 nm LED light source. The printing process was performed with a 200 μm layer thickness, an approach and separation velocity of 0.2 mm/min, a slider velocity of 1 mm/min, and a 3 s wait time after each step. The light intensity was set at 21 mW/cm² with a burn-in irradiation time of 8 s for two layers and a normal exposure time of 5 s for the remaining layers. The entire printing process occurred under a water aerosol flux between 10 and 30 ml/h generated by a TaoTronics TT-AH002 humidifier. After printing, the samples were cleaned using compressed air to remove the unpolymerized resin, and for some structures that have residual resin, it was manually removed with a cloth wetted with ethanol. The objects were then post-cured after removing the unreacted resin to complete the polymerization and to strengthen the printed objects. Post-curing was performed in a UV chamber with a medium-pressure mercury lamp (Asiga Flash or RobotFactory UV oven) for 2 min.

Printed samples were stored in a sealed environment with a high moisture content to prevent excessive water evaporation. The resolution was estimated by the digitalization of the photographs of 3D objects designed specifically for this purpose (Supplementary Fig. 8). This CAD file was printed to determine which was the finest feature that could be printed.

**Formulation characterization**. Fourier Transform Infrared Spectroscopy (FT-IR) tests were conducted on Nicolet iS FTIR 50 Spectrometer (Thermo Fisher, Germany) using Attenuated Total Reflection (ATR) mode with a Smart iTX module collecting 32 scans from 400 to 4000 cm$^{-1}$. Rheological measurements of formulations with increasing PVA concentrations were performed using a rheometer (Anton Paar Physica MCR 302) in a 25 mm diameter parallel plate mode. The gap between the two plates was set to 0.2 mm, and the formulation was kept at a constant temperature of 25 °C. Amplitude sweep tests were performed at a constant frequency of 10 rad/s over a strain ramp ranging from 0.01 to 1000%. The viscosity of the inks was tested in a rotation shear ramp test ranging from 0.01 1/s to 1000 1/s over 2 min. Real-time photo-rheology tests were performed with a setup comprising a quartz lower plate and under a constant shear frequency of 10 rad/s and a constant strain amplitude of 1% in the linear viscoelastic region. The light source used was a Hamamatsu LC8 lamp, equipped with an 8 mm light guide, with a bulb emitting UV light with an intensity on the quartz window between 23 and 25 mW/cm² (measured with a UV radiometer EIT Power Puck II). The light was turned on after 60 s to stabilize the system. Concomitant changes in viscoelastic material moduli during polymerization were measured as a function of exposure time.

**SH mechanical characterization**. SH experiments were conducted by cutting the samples, printed in the shape of dog bones, into two equal pieces with a sharp blade to separate the parts. The cut faces were then brought in contact with each other and gently pressed together and held in contact by hand for few seconds to ensure uniform surface contact without any other external force. Rejoined samples were left to heal at room temperature for various allotted times, from 1 to 12 h, stored inside closed and sealed vessels with saturated moisture to decrease water evaporation. The mending efficiency was quantified, subjecting both pristine and rejoined samples to uniaxial tensile tests. Mechanical properties were measured using Universal Testing Machine (Instron 3345 or MTS QTest/10, both equipped with a 10 N load cell) testing printed dog-bone samples. The dumbbell-shaped specimens were inspired by a standard sample (ASTM D1708–18) with a length of 40 mm, a gauge length of 22 mm, a width of 15 mm, a gauge width of 5 mm, and a thickness of 3 mm. The test was conducted at room temperature under a crosshead speed of 50 mm/min and a sample length between the jaws of 12 ± 1 mm, recording the load and elongation at the peak of the curve. The nominal stress was calculated as the force over the initial cross-section area of the sample, while the strain was calculated as the percent of the ratio of the final length at the break over the initial sample length between jaws. A minimum of five samples for each batch was tested, and the results were averaged for reproducibility. Samples were held between knurled clamps or clamps covered with anti-slip strips to secure the slippery hydrogels. Bulk UV irradiation of the formulation was performed, pouring the formulation in a dumbbell-shaped silicon mold and placing it under a medium pressure mercury lamp for 2 min (Asiga Flash). Both sets of samples were tested under identical conditions.

## Data availability

All the relevant data of this study, including raw data, are available within the manuscript, in the supplementary information file or from the authors, by request.

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

## Acknowledgements
This work was supported by Compagnia di San Paolo through the funding scheme "Joint Project with Top Universities". Financial support was also provided from DEFLeCT project funded by Regione Piemonte POR-FESR 2014–2020. The authors would like to thank also Gaia Di Martino for her technical support.

## Author contributions
S.M. and I.R. proposed the concept of this work and directed the project. M.C. designed the research and carried out the experiments. A.C. helped with the rheological characterization. L.L. provided illuminating discussions. M.C. drafted the manuscript, and M.C., S.M., and I.R. contributed to the writing of the manuscript and supporting information, C.F.P. supervised the project.

## Competing interests
The authors declare no competing interests.
