## [Peer Review File · Nature Communications]

REVIEWER COMMENTS

Reviewer #1 (Remarks to the Author):

This manuscript reports a water-soluble ink compatible with stereolithography (SLA) to fabricate self-healable, 3D hydrogel objects. The ink was developed by mixing commercially available materials including poly (ethylene glycol) diacrylate (PEGDA), acrylic acid (AAc), and poly (vinyl alcohol) (PVA). Upon photo-crosslinking, the solidified ink features a double network system, where the chemically-crosslinked network consisting of PEGDA and AAc primarily provided mechanical integrity, and the physically-crosslinked network containing PVA imparted healing capability due to hydrogen bonding. The optimal formulation with a PVA/AAc ratio of 0.8 exhibited the best self-healing efficiency within the printing window, and autonomous healing occurred instantly in a high humidity environment. Although this work reports the fabrication of self-healing hydrogel using SLA, it demonstrated inadequate chemistry novelty since the ink is a simple blend of commercially available materials, all of which had been extensively reported previously (i.e. PEGDA and AAc had been widely utilized for SLA to provide high-resolution objects, and PVA is a common material to produce self-healable hydrogels). In addition, the printing resolution, mechanical properties, and self-healing ability of the objects reported in this paper are not dramatically different from the prior literature, given the widely reported excellent properties of the above-mentioned materials. Therefore, this manuscript is not suitable for publication in Nature Communications due to the limited impact of this paper upon the 3D printing field and materials design field.

- The authors should explain why the resolution along z-axis is strongly affected by the ink viscosity while the resolution in xy plane is related to dye concentration.
- The authors must provide the healing efficiency of the object in the sealed environment without the application of moisture. The authors indicated the object was able to self-heal instantly without any external trigger. However, this is incorrect since humidity is a very important and effective stimulus for self-healing, especially since PVA is water-soluble. In addition, the printed material itself already contained a large proportion of water and should provide PVA high mobility to migrate across the ruptured interface.
- The author must provide convincing mechanical property data and a more detailed explanation to elucidate the self-healing performance of the printed sample. The stress-strain data shown in Figures 4a and 4b demonstrated no obvious trend and large variance. In addition, the self-healing efficiency (% of strength or strain recovered) is still pretty low, given the large water content, high PVA content, and low mechanical strength of the printed samples.
- The authors should consider submission to an applied polymer chemistry oriented journal once the healing properties are more clearly defined in a quantitative manner

Reviewer #2 (Remarks to the Author):

[Note from the editor: See also remarks in the attachment]

This is a very interesting and timely article. In general it is of a high quality and has novelty, and will be of interest to those in additive manufacturing research and also outside of AM where the structures manufactured by this technology could have benefit (as outlined by the authors). I have identified a number of omissions in the paper and these are highlighted in the reviewed document and summarised below:

1. The challenges of conventional hydrogel printing are not well described and need fully details in the introduction.
2. The suitability of DN hydrogels to 3D bioprinting and extrusion printing is not obvious and needs to be explained more fully.
3. There are some inaccuracies in the introduction. In particular, LN 64 P3, I would argue that Material Deposition techniques are more capable of providing higher strength than VP approaches as thermoplastics and highly filled materials may be used. Re-write this to be more accurate.
4. You describe that you are using SLA printing, but the DLP process is not SLA, it is Vat Polymerisation, but not SLA. This requires correction.
5. The polymer is referred to as an interpenetrating system in a number of places throughout, and importantly, in one of the conclusions, although there is no scientific evidence or data to substantiate that this system is a true IPN. You need to prove that it is an IPN or remove IPN from the paper.
6. Resolution is mentioned but there is no description of how resolution was measured? It is not clear in Figure S5 how the resolution is determined and how the xy and z resolutions are measured. The method of quality assessment and resolution should be in the methods section.
7. The self-healing mechanism is proposed as formation of H-bonds but there is no data or evidence to support this. The authors need to state how they know this and what data shows that this is the mechanism (otherwise it is an unproven theory). Could / was UV-VIS or FT-IR be used to look for chemical bond changes?
8. How does Fig S6 show repeatable restoration. The stress and strain values are different for each cycle, so it is not repeatable. Authors need to explain this better.
9. The statement is made 'This approach enabled reaching a complexity of architecture and a resolution not 195 achievable with conventional extrusion and bioprinting printing processes'. This is not conclusive as you do not provide the resolution or complexity information for the conventional extrusion or bioprinting processes (and do not describe them). Need to add details of these processes in the lit review with their resolution and complexity (however that is measured) capabilities. A table here to compare the capabilities of the three manufacturing processes would be useful and should be provided.

Reviewer #3 (Remarks to the Author):

This manuscript describes a very interest work and it could be a breakthrough in the knowledge of the field. The construction of 3D-self-healing hydrogels is, as far as I know, not described up to now. In my opinion the work is really novel, well designed and it broadens the field to new devices with complex architecture. Conclusions are adequate and I feel especially important the fact that the self-healing process takes place in the absence of external stimuli. I recommend its publication after a major revision.

I have some concerns about the actual state of the manuscript. There are some mistakes, non-explained results and some assesments that need experimental support or justification. These details need a careful major revision. In addition, english and some typewriting mistakes should be carefully corrected.

-L17, Abstract: is the photoinitiator commercially available? If not (as it is described in the Materials and Methods section), please change the sentence.

-L21, Abstract: "recovers after 10 hrs". In the rest of the manuscript is said "after 12 hrs". Please, unify time.

-L94, System Design: PVA polymer of a single molecular weight is used. The crucial feature of the system is related to: a) viscosity of initial formulation, and b) PVA chain diffusion and interpenetration over the interfaces, and both parameters strongly depend on PVA Mw. Authors should check other MWs or justify why they are not using them.

-L103, System Design: "application of our newly-developed water-compatible photoinitiator nanoparticles was crucial to obtain rapid printing". This assumption should be justified or supported by comparison with experimental results obtained using an efficient water soluble photoinitiator.

L107: Authors should justify why they say that PVA chains are homogeneously distributed.

-L141: "the high monomer reactivity caused an over-curing of the object together with the thickening of the ink". Please explain why overcuring is a problem and, at the same time, you need a post-curing step.

-L162-164 and L171: I find very elegant the production of samples with different colors, as healing could be then easily seen. But the addition of the dye has an essential impact in the photopolymerization reaction (penetration of the light, resolution...). Other features of photopolymerization (especially conversion) should be affected when dye is changed. The different conversion/crosslinking density could cause that migration of dye is only observed in one direction (red dye to green area), and purple color is only seen at one side of the healed cut (Fig 3c-inset)(and so PVA chains should do). Change of dyes is only done for a better view of the healing area, but I think that authors should explain this.

- Materials and methods: authors use three different irradiation systems (385nm led, non-filtered Hamamatsu UV lamp, and medium pressure Hg lamp). Please justify why some experimental values obtained from one are used to fix experimental conditions in other one. And why conclusions obtained with Hg irradiation systems could be extrapolated to 385nm-led irradiation one.

-L245: "All the printed samples were rapidly cleaned with a cloth wetted with ethanol" How these complicated shapes could be cleaned with a wetted cloth?

Paula Bosch

Reviewer #1 (Remarks to the Author):

This manuscript reports a water-soluble ink compatible with stereolithography (SLA) to fabricate self-healable, 3D hydrogel objects. The ink was developed by mixing commercially available materials, including poly (ethylene glycol) diacrylate (PEGDA), acrylic acid (AAc), and poly (vinyl alcohol) (PVA). Upon photo-crosslinking, the solidified ink features a double network system, where the chemically-crosslinked network consisting of PEGDA and AAc primarily provided mechanical integrity, and the physically-crosslinked network containing PVA imparted healing capability due to hydrogen bonding. The optimal formulation with a PVA/AAc ratio of 0.8 exhibited the best self-healing efficiency within the printing window, and autonomous healing occurred instantly in a high humidity environment. Although this work reports the fabrication of self-healing hydrogel using SLA, it demonstrated inadequate chemistry novelty since the ink is a simple blend of commercially available materials, all of which had been extensively reported previously (i.e. PEGDA and AAc had been widely utilized for SLA to provide high-resolution objects, and PVA is a common material to produce self-healable hydrogels). In addition, the printing resolution, mechanical properties, and self-healing ability of the objects reported in this paper are not dramatically different from the prior literature, given the widely reported excellent properties of the above-mentioned materials. Therefore, this manuscript is not suitable for publication in Nature Communications due to the limited impact of this paper upon the 3D printing field and materials design field.

We would like to thank the reviewer for her/his valuable comments and hope that the revised manuscript, which addresses the reviewer's comments, will provide convincing arguments for the suitability of the manuscript for publication in Nature Communications.

In this paper, as correctly pointed by the reviewer, we did not synthesize new materials. On the contrary, we use commercially available materials in such compositions that can enable self-healing of 3D printed objects. As we stated in the introduction, fabrication of 3D self-healing hydrogels by photopolymerization based printing technologies was not reported until now, although light-activated technologies offer many advantages compared to other methods such as Direct Ink Writing. We believe that achieving this goal is a significant step forward in this field. By using commercially available materials, our research would enable fast adaptation of the approach to researchers from a variety of fields, including bioprinting. For example, since we used irradiation at 405 nm, this approach can potentially be very suitable for the emerging field of 3D printed organs since the cells will not be affected by the printing process.

Note that in a very recent review, this need for simpler processes and friendly materials is also emphasized (Zhu, M., et al., Eur. Polym. J. 129, 109651 (2020), now Ref. 56), as follows: "These processes involve unavoidable shortcomings such as stringent reaction conditions, the use of toxic chemicals and complex synthesis and purification procedures. Complexity hinders the mass production and economic practicability and further restricts its wide application". Therefore, we added this reference to the conclusion, along with the following sentence:

L236: "The complexity related to stringent reaction conditions and complex synthesis and purification procedures for self-healing materials hinders mass production and economic practicability and, together with the difficulties related to the shaping, the potential applications are restricted⁵⁶. The processing of commercially available materials via a widely available apparatus to fabricate 3D printed hydrogels would enable fast adaptation of the proposed approach to open the way for new applications in a variety of fields, ranging from biology^{2,23,34}, and implantable sensors⁵⁷ to soft robotics^{58,59} and energy storage⁶⁰"

Therefore, we believe that this work, which bridges different disciplines, could be of interest to the broad readership of Nature Communication.

1.1 The authors should explain why the resolution along z-axis is strongly affected by the ink viscosity while the resolution in xy plane is related to dye concentration.

The resolution is affected by both the dye and the properties of the printing composition, including viscosity and polymerization rate. The presence of a light absorber helps the resolution in the 3D printing process due to the confinement of radiation penetration and better control of the photopolymerization kinetics. The addition of a light absorber resulted in a better control of the XY resolution because curing outside the irradiated areas is avoided. This phenomenon is well reported and discussed in detail in the review paper in ref 34 in the manuscript. For the z-resolution, we have found that the increase in viscosity led to delamination of the thin layer, in such case, we had to increase the layer thickness, meaning decreased z-resolution, to avoid failure of the printing process.

In order to demonstrate the positive effect of the presence of the dye on the printing process, we have now added a figure in the supporting information file to support our observations (Supplementary Figure 1). We printed cuboid samples (25 x 10 x 5 mm) with formulations containing two different dyes, methyl red (left) and brilliant green (right), and we compared them to a sample printed without adding a dye (same STL printing file). As seen, the sample printed with no dye shows extensive curing, and the shape is undefined, while the addition of a dye results in good shape fidelity with sharp edges and flat surfaces.

To clarify the resolution issue, we have changed the “results” section as follows:

L156: “In addition, to avoid polymerization over the exposed areas and to achieve a better resolution, a water-soluble dye was added, leading to better confinement of the polymerization at the x-y plane, lower light penetration depth, and more controlled photoinitiation kinetics^{7,34,55}. To show this effect, cuboid samples (25 x 10 x 5 mm, same STL printing file) were printed with formulations containing two different dyes, methyl red (left) and brilliant green (right), and then compared to a sample printed without adding a dye (Supplementary Figure 1). The sample printed with no dye shows extensive polymerization and its shape is undefined, while the addition of a dye results in good shape fidelity with sharp edges and flat surfaces. As demonstrated in Supplementary Figure 1, the presence of the dye improves the resolution in the XY plane, providing a better final quality of the printed objects⁵⁵. At the same time, the resolution along the Z-axis is not only related to the dye concentration but is also strongly affected by the ink viscosity

(and, therefore, by the PVA concentration) that results in a tendency to delaminate the layers immersed in a thick formulation because of an increase in the adhesion forces (Supplementary Figure 6).”

1.2 The authors must provide the healing efficiency of the object in the sealed environment without the application of moisture. The authors indicated the object was able to self-heal instantly without any external trigger. However, this is incorrect since humidity is a very important and effective stimulus for self-healing, especially since PVA is water-soluble. In addition, the printed material itself already contained a large proportion of water and should provide PVA high mobility to migrate across the ruptured interface.

We agree with the reviewer that there are examples of water-triggered self-healing in dry systems, but in these systems, the water is added locally at the cut interfaces (see, for example, Ref. 21). In our work, the printed objects are composed of a hydrogel, in which water is a necessary and major component of the structure, and therefore, in our view, water cannot be considered as an external stimulus, as also described in the review on self-healing hydrogels (Ref. 36). Furthermore, the designed complex-shaped hydrogel showed effective adhesion immediately after rejoining (L178) without any external assistance or stimulus, except for the manual contact of the interfaces. However, the completion of the self-healing process was achieved only over time, about 12 hours (L190).

According to the reviewer’s suggestion, we performed the suggested experiment, keeping the samples in a sealed container without the addition of humidity. Dumbbell-shaped specimens were 3D printed and tested in tensile tests. Self-healing properties were evaluated by testing the samples every 24 hours for five consecutive days, rejoining the samples after rupture, and keeping them in a sealed environment. The results are reported in the left figure below, which has also been added to the SI file as Supplementary Figure 9b. As can be seen, the restoration took place every time with lower efficiency, while the first healing process was similar to that obtained for the samples stored under humidity. This behavior can be explained considering the minor loss of weight that we measured during the time, as reported in the figure below on the right, also added in the SI file as Supplementary Figure 9c. This weight loss, unfortunately, could not be avoided because of water evaporation during handling and testing. These results confirm the need to keep a constant amount of water in the hydrogel to preserve the SH properties over time.

In order to clarify this point, we modified the text as follows:

L116: “Furthermore, as will be shown, the self-repair occurs without external stimuli or adhesives and is only due to internal forces because the printed material itself already contains a large fraction of water, providing to PVA high mobility to migrate across the ruptured interface.^{14,45}”

L205: “The samples were kept in a sealed closed vessel with a humid environment to reduce water evaporation and maintain controlled conditions after-printing during the entire healing process.”

L208: “A second set of experiments was performed keeping the samples in a sealed container without controlled humidity, evidencing a lower recovery efficiency every cycle (Supplementary Figure 9b). This behavior can be explained by the loss of water during setting up the sample to measure the self-healing (Supplementary Figure 9c). As expected, keeping a constant water concentration in the hydrogel is crucial to preserve the SH properties.”

1.3 The author must provide convincing mechanical property data and a more detailed explanation to elucidate the self-healing performance of the printed sample. The stress-strain data shown in Figures 4a and 4b demonstrated no obvious trend and large variance. In addition, the self-healing efficiency (% of strength or strain recovered) is still pretty low, given the large water content, high PVA content, and low mechanical strength of the printed samples.

We want to thank the reviewer for stressing this aspect of the study. However, there is no commonly shared standard for both self-healing procedure and testing. In literature, many studies are reporting on procedures for evaluating self-healing that are not totally comparable (in the manuscript, see Ref. 28-30, 36, 40, 45-48). In order to compare the performance, we mainly refer to the following studies on similar systems:

Ref. 45 10.1021/mz300451r (2012) hydrogel based on neat PVA, self-healing: 72%

Ref. 46 10.1021/acs.langmuir.5b03474 (2015) Hydrogel based on PEGDA/PVA, self healing: 68%

Ref. 47 10.1021/acsami.6b05627 (2016) Hydrogel based on PVA/Acrylamide and AAc, self-healing 37%

Ref. 48 10.1016/j.cej.2019.04.077 (2019) hydrogel based on PVA/PAAc ionically cross-linked, self-healing: 100%

We obtained a self-healing efficiency of 72%, which can be considered good given the literature. It should be noted that in our study, we had to face not only the problem of restoration but also the limitation arising from the fabrication of complex shapes. In our opinion, the data presented in this work are remarkable, given the mitigation of the H-bonds interactions due to the presence of a high amount of water.

It should also be noted that the value of 100% presented in Ref. 48, was obtained thanks to the combination of hydrogen bonding with a second SH mechanism, based on the addition of ferric ions. This approach is indeed interesting, and it will be evaluated in our future investigations.

We agree with the reviewer about the data variance; however, we would like to point out that this can be related to the manual operation to carry out the self-healing procedure and testing. Unlike typical simple shapes in SH studies, we could not provide an exact alignment of the rejoined surfaces because of the small dimensions during the manual attachment. Moreover, the swollen nature of the hydrogel caused deformation upon compression while clamping the samples in the tensile test machine, with a possible

misalignment among the clamps. Lastly, the experimental variation may also be attributed to the irregular and gradual rupture of the healed samples at the contact interface. Nevertheless, the general trend is visible with the top performances achieved after 10-12 hours in contact.

To support the good agreement of our results with the literature, we added the following sentence:

L198: “Printed dog-bone samples (0.8 weight ratio PVA/AAC) showed an average of 72% recovery in tensile after healing for 12 hours, in good agreement with the results reported for neat PVA-based non 3D-printed hydrogels, in which the addition of a chemically cross-linked network had detrimental effects on the self-healing^{45,46}”

- The authors should consider submission to an applied polymer chemistry oriented journal once the healing properties are more clearly defined in a quantitative manner

As mentioned above, we are sorry that the reviewer did not share our view on the novelty of our system in the field and its expected impact on the field. We hope that after the modifications of the manuscript following the reviewers' comments, the reviewer could share this opinion as well. We expect that this investigation may open new opportunities and expand the range of application of additive manufacturing, and so it could be of interest for the broad readership of Nature Communication.

Reviewer #2 (Remarks to the Author):

This is a very interesting and timely article. In general it is of a high quality and has novelty, and will be of interest to those in additive manufacturing research and also outside of AM where the structures manufactured by this technology could have benefit (as outlined by the authors). I have identified a number of omissions in the paper and these are highlighted in the reviewed document and summarised below:

We would like to thank the reviewer for her/his nice evaluation of our work and the constructive comments. The replies to these comments are as follows:

2.1 The challenges of conventional hydrogel printing are not well described and need fully details in the introduction.

We modified the text in order to highlight the challenges of hydrogel printing. The introduction was revised as follows.

L36: “This technology enables overcoming many of the challenges of conventional hydrogel printing, such as the crucial need for high viscosity of the printing composition, in order to facilitate sufficient fixation of the printed structure, and the need to use supports to print hollow parts, depending on their orientation^{5,6}. In VP, the projection of light triggers a localized solidification of a liquid formulation, layer-by-layer, leading to the fabrication of precise 3D structures with fine spatiotemporal control, without the need for supporting structures or additional materials⁷⁻⁹.”

2.2 The suitability of DN hydrogels to 3D bioprinting and extrusion printing is not obvious and needs to be explained more fully.

We thank the reviewer for the suggestion. We modified the introduction accordingly:

L53: “It has been shown that IPN hydrogels match extrusion-based 3D printing material requirements^{5,25,26}. The viscosity of the ink, given by macromolecules interaction, can be tuned to be suitable for extrusion, while the chemical network could rapidly “fix” the shape^{27,28}. The physical network was also exploited to impart self-repairing characteristics, and this approach enabled the fabrication of self-healable 3D constructs with very simple shapes²⁸⁻³⁰.”

2.3 There are some inaccuracies in the introduction. In particular, LN 64 P3, I would argue that Material Deposition techniques are more capable of providing higher strength than VP approaches as thermoplastics and highly filled materials may be used. Re-write this to be more accurate.

We agree with the reviewer, and we modified the text accordingly:

L57: “Material extrusion-based 3D printing technologies for hydrogels are the most common in the field of biofabrication and provided an incredible step forward in the development of customized replicas, mimicking natural structures with controlled geometry and characteristics³¹

L60: “In addition, these technologies are limited by the high ink viscosity and structural deformations or failures^{11,32}.”

L66: “In vat polymerization, unlike the extrusion-based process, there is no need for support material, and therefore lattices with overhanging features and through-holes show clean and sharp edges¹¹. Moreover, there are fewer limitations on the suitable viscosity range for the fabrication process, but the material choices are limited to liquid or soluble photopolymers, which might be a disadvantage^{5,34}. Low viscosity allows increasing the amount of solvent in the inks up to 90% in volume without compromising the final precision³⁵.”

2.4 You describe that you are using SLA printing, but the DLP process is not SLA, it is Vat Polymerisation, but not SLA. This requires correction. (in the PDF file from the reviewer we were asked to add a scheme and a definition)

We agree with the reviewer, and we corrected the manuscript accordingly. We used SLA as a generic term since it is the most commonly used term in commercial printers, while indeed vat photopolymerization is the correct term for the class of 3D printing techniques based on a liquid resin contained in a tank. We also added a definition of DLP processing and the scheme of our printing system, as suggested.

L121: (Results-DLP 3D printing) “Digital light processing (DLP) is a sequential layer-by-layer bottom-up maskless photolithography technique. An entire slice of the object with controlled thickness is selectively solidified in a single exposure by a UV or visible light projection of a 2D pattern on a liquid photocurable resin. The build platform is then moved to fill the printing area with the uncured resin before fabricating the next layer, repeating the process until complete fabrication. A schematic representation of the DLP printer in our setup is shown in Fig. 2a.”

We also added a schematic representation of the DLP printing apparatus used in the work as Fig. 2a, also reported below:

2.5 The polymer is referred to as an interpenetrating system in a number of places throughout, and importantly, in one of the conclusions, although there is no scientific evidence or data to substantiate that this system is a true IPN. You need to prove that it is an IPN or remove IPN from the paper.

Combining two different polymers that are not covalently bound together in one system was reported and recognized as interpenetrated networks (Dragan, E. S. Design and applications of interpenetrating

polymer network hydrogels: A review, Chem. Eng. J. 243, 572–590 (2014), now added as Ref. 24). Interpenetrating network hydrogels (IPNs) are composed of separate networks physically entangled within each other without any covalent bonds between them, with at least one of them being cross-linked within the immediate presence of the other. According to the definition reported in the reference mentioned above, our system is a sequential semi-IPN since the precursors of the chemical covalent network are added to a linear polymer solution that is entrapped in the cross-linked matrix after polymerization. We called the system “double network” to highlight the different nature of the constituents, but in compliance with the specific definition, it was not completely consistent. Therefore, we changed the term “double network” throughout the manuscript to semi-IPN.

We modified the text accordingly:

L49: “A convenient strategy to gather 3D shaping (printability) and self-healing properties consists in designing hydrogels with an interpenetrated network (IPN)^{21,22}. Those networks combine a rigid and robust frame (chemical bonds, generally non-reversible) with a much weaker one (physical bonds, mostly reversible)²³, displaying new tailored features, such as improved toughness and flexibility while retaining the characteristics of both constituents simultaneously²⁴. It has been shown that IPN hydrogels match extrusion-based 3D printing material requirements^{5,25,26}.”

L97: (Results-System design) “The system was designed as a sequential semi-IPN since the precursors of the chemical covalent network are added to a solution containing a linear polymer that is entrapped in the cross-linked matrix after polymerization²⁴.”

Regarding the tests, it is impossible to have direct evidence of the configuration of the network in the wet hydrogel state. We have now reported in SI (Supplementary Figure 2) the ATR-IR spectra collected in different points of dried samples in which both typical peaks of PVA and AAc are present simultaneously (C=O stretching of AAc at 1708 cm⁻¹ and OH rocking of PVA at 1325 cm⁻¹). This result, together with the transparency of the formulations and the final samples, allows concluding that there are no separate phases, and so the two networks can be considered homogeneously mixed.

The following sentence was added to the revised manuscript:

L112: “The homogeneity is confirmed both by the transparency of the formulations and printed objects and by ATR-IR spectra collected in different points of dried samples, in which typical peaks of both PVA and AAc are present simultaneously (Supplementary Figure 2).”

2.6 Resolution is mentioned but there is no description of how resolution was measured? It is not clear in Figure S5 how the resolution is determined and how the xy and z resolutions are measured. The method of quality assessment and resolution should be in the methods section.

Thank you for your comment. The resolution was estimated by the digitization of the photographs of 3D objects explicitly designed for this purpose. The CAD file reported below is added as a new figure in Supplementary Figure 7. This CAD file was printed to determine which was the finest feature that we could print.

The following sentence was added to the experimental section:

L273: “The resolution was estimated by the digitalization of the photographs of 3D objects designed specifically for this purpose (Supplementary Figure 7). This CAD file was printed to determine which was the finest feature that could be printed.”

2.7 The self-healing mechanism is proposed as formation of H-bonds but there is no data or evidence to support this. The authors need to state how they know this and what data shows that this is the mechanism (otherwise it is an unproven theory). Could / was UV-VIS or FT-IR be used to look for chemical bond changes?

We are sorry for not having made clear the basis behind our statements. In line 97 we previously wrote

“PVA was selected since it is biocompatible and possesses abundant hydroxyl groups, thus providing efficient self-healing that is enabled by extensive hydrogen bonding.”

Indeed, the hydrogen bonding mechanism is commonly accepted (ref 45-48), but to the best of our knowledge, it is still a hypothesis, and therefore we slightly modified the text in line 103 to indicate that this is a hypothesis.

L103: “PVA was selected since it is biocompatible, possesses abundant hydroxyl groups able to form extensive hydrogen bonding, and is a well-known material to produce efficient SH hydrogels^{45-48,50,51}.”

The strong similarity between hydrogen bonding among PVA chains and the H-bonds in water makes it impossible to analyze the formation of interchain interactions across the ruptured surfaces. This issue was reported, for example, by Zhang H. *et al.* in Ref. 45, who performed FTIR studies but could not reveal direct support for the H bonds by the PVA while in water. In our system, the bands of hydroxyl groups of PVA chains are also hidden by the dominant bands of water molecules (65 wt% water in the hydrogel).

Therefore, we revised the conclusion part in the manuscript to indicate that the H-bonding is a hypothesis that should be proven in the future while overcoming the problem of distinguishing such bonds in systems containing very high water concentrations. The following sentence was added:

L228: “At this stage, the H-bonding mechanism is a hypothesis that should be proven in the future, overcoming the problem of distinguishing such bonds in systems containing a very high water concentration.”

2.8 How does Fig S6 show repeatable restoration. The stress and strain values are different for each cycle, so it is not repeatable. Authors need to explain this better.

We replaced Fig. S6 with a new figure, now Supplementary Figure 9a, while including additional experimental results. Now we include the data for the pristine uncut sample and modified the graph to include the self-healing efficiency percentage. It is possible to observe that the repeated repaired sample showed an average healing of about 65% with certain repeatability that falls in the error range. The variability in the results can be mostly attributed to the manual reattachment of the samples, which could not always be rejoined in the same exact way. However, we performed the tests on multiple specimens to reduce the experimental uncertainty.

The manuscript was revised accordingly, as follows:

L205: “The samples were kept in a sealed closed vessel with a humid environment to reduce water evaporation and to maintain controlled conditions after-printing during the entire healing process. This was crucial to preserve the SH properties, as also demonstrated by control experiments. It should also be noted that the objects possess repeatable mechanical restoration with average healing of about 65% with repeatability that falls in the error range (Supplementary Figure 9a). A second set of experiments was performed keeping the samples in a sealed container without controlled humidity, evidencing a lower recovery efficiency every cycle (Supplementary Figure 9b). This behavior can be explained by the loss of water during setting up the sample to measure the self-healing (Supplementary Figure 9c). As expected, keeping a constant water concentration in the hydrogel is crucial to preserve the SH properties.”

2.9 The statement is made ‘This approach enabled reaching a complexity of architecture and a resolution not 195 achievable with conventional extrusion and bioprinting printing processes’. This is not conclusive as you do not provide the resolution or complexity information for the conventional extrusion or bioprinting processes (and do not describe them). Need to add details of these processes in the lit review with their resolution and complexity (however that is measured) capabilities. A table here to compare the capabilities of the three manufacturing processes would be useful and should be provided.

To the best of our knowledge, there is no quantitative comparison of “complexity” in the literature, but a visual comparison of similar objects fabricated via the different processes can be performed. To illustrate this issue, below we present images that were taken from different studies (Ref. 29,49,52). As can be seen, extrusion printing (top left and bottom left) shows warped and slightly deformed planes due to the shape of the extruded materials upon fixation, which is usually based on rheology or solidification of the

material. On the other hand, vat photopolymerization (top right and bottom right) enables obtaining flat vertical surfaces with negligible distortion and great shape fidelity because the conversion of the liquid into a solid is very fast since it is based on a photopolymerization reaction. In addition, in Vat polymerization, unlike the extrusion-based processing, there is no need for support material, and therefore lattices with overhanging features and through-holes show clean and sharp edges.

Freeform vertical structures with no overhanging features and cavities fabricated with extrusion printing (left) and vat photopolymerization (right). (Ref. 29,52)

Simple cubic lattices fabricated with extrusion printing (left) and vat photopolymerization (right). (Ref. 29,49)

As suggested by the reviewer, we added a paragraph to address this issue in the introduction, along with a table and the related references in the supplementary information (Supplementary Table 1), as presented below, while comparing the characteristics of the different printing processes.

Table: Characteristics comparison of extrusion printing and vat photopolymerization

Process	Material	X-Y Resolution [1]	Z Resolution [2]	Model Surface [2]
Material Extrusion (FDM,FFF,DIW, Bioplotter)	Viscous Liquids Thermosoftening polymers	100–200 μm	100–200 μm	Very rough
Vat Photopolymerization (SLA, DLP, CLIP)	Liquid photocurable resins	SLA - 1–10 μm DLP - 20–200 μm	> 1 μm	Smooth

Process	Material Deposition [3]	Fabrication Speed [4]	RTM Ratio [5]	Viscosity [6,7]
Material Extrusion (FDM, FFF, DIW, Bioplotter)	Continuous line	Slow (μm/s)	0.5–1 m ² /min	10 ⁵ –10 ⁷ mPa·s
Vat Photopolymerization (SLA, DLP, CLIP)	SLA - line rastering DLP - layer exposure	SLA - medium (mm ² /s) DLP - fast (mm ³ /s)	0.5–2 m ² /min	10 ² –10 ⁴ mPa·s

The manuscript was, therefore, revised as follows:

L59: “Typically, extrusion printing shows warped and slightly deformed planes due to the shape of the extruded materials upon fixation, usually based on rheology or solidification of the material¹¹.”

L61: “Extrusion-based printing also generally operates at low building speed and resolution (Supplementary Table 1)³³. On the contrary, VP is capable of fabricating 3D hydrogels with higher geometrical complexity and finer accuracy, with no substantial spatial resolution effect on the printing time^{10,11,33}. It enables obtaining flat vertical surfaces with negligible distortion and great shape fidelity because the conversion of the liquid into a solid is very fast since it is based on a photopolymerization reaction. In addition, in vat polymerization, unlike the extrusion-based process, there is no need for support material, and therefore lattices with overhanging features and through-holes show clean and sharp edges¹¹.”

We also modified the conclusion in the main manuscript as follows:

L215: “This approach enabled reaching a complexity of architecture, especially in terms of overhanging and hollow features with sharp edges, which are not achievable with conventional extrusion-based printing processes.”

Reviewer #3 (Remarks to the Author):

This manuscript describes a very interesting work and it could be a breakthrough in the knowledge of the field. The construction of 3D-self-healing hydrogels is, as far as I know, not described up to now. In my opinion the work is really novel, well designed and it broadens the field to new devices with complex architecture. Conclusions are adequate and I feel especially important the fact that the self-healing process takes place in the absence of external stimuli. I recommend its publication after a major revision.

I have some concerns about the actual state of the manuscript. There are some mistakes, non-explained results and some assessments that need experimental support or justification. These details need a careful major revision. In addition, English and some typewriting mistakes should be carefully corrected.

We would like to thank the reviewer for the evaluation of the work and the useful comments. We revised the manuscript, and we hope the reviewer will find the paper improved by our corrections based on the suggestions.

3.1-L17, Abstract: is the photoinitiator commercially available? If not (as it is described in the Materials and Methods section), please change the sentence.

We updated the text with the commercial name as can be found on Sigma-Aldrich's website (<https://www.sigmaaldrich.com/catalog/product/aldrich/906808>):

L244: "Water-soluble TPO based nanoparticle photoinitiator (containing ionic surfactant)"

3.2-L21, Abstract: "recovers after 10 hrs". In the rest of the manuscript is said "after 12 hrs". Please, unify time.

We corrected the inconsistency.

3.3-L94, System Design: PVA polymer of a single molecular weight is used. The crucial feature of the system is related to: a) viscosity of initial formulation, and b) PVA chain diffusion and interpenetration over the interfaces, and both parameters strongly depend on PVA Mw. Authors should check other MWs or justify why they are not using them.

Thank you for this comment. Indeed, we performed many preliminary studies to determine the most suitable Mw and concentration of the PVA to match DLP printability and efficient self-healing. For the sake of clarity, we did not include this information in the submitted manuscript.

In view of the comment, we have revised the manuscript as follows:

L261: "Preliminary experiments were performed to evaluate the optimal Mw of the PVA (Supplementary Figure 10), and based on these experiments, the Mw of 89,000-98,000 was selected"

In Supplementary Figure 10, it can be deduced that lower Mw did not show adequate self-healing even at a high weight percentage, 30% in water, the highest achievable that allow suitable viscosity for DLP. Likewise, higher Mw did not show acceptable efficiency, resulting from the excessive viscosity even at low concentrations.

3.4-L103, System Design: “application of our newly-developed water-compatible photoinitiator nanoparticles was crucial to obtain rapid printing”. This assumption should be justified or supported by comparison with experimental results obtained using an efficient water soluble photoinitiator.

We needed to use a PI active in the wavelength range of our 3D printers (385 nm), so we could not use the very common water-soluble photoinitiator I2959, limited in absorption to 365 nm. The most suitable in that range of interest are acylophosphine-based species (TPO and BAPO). However, those PIs have very low water solubility, and therefore they are not suitable for being used in our system. The PI here employed was developed in our laboratories, so we know that it is highly efficient for waterborne formulations (Ref. 49,52)

3.5 L107: Authors should justify why they say that PVA chains are homogeneously distributed.

This point was also addressed in reply to reviewer 2, as follows: Our claim about homogeneity is based on observations made on the transparency of the liquid formulation and the printed samples. No light diffraction can be seen, so it is possible to conclude no phase separation. Regarding the tests, it is impossible to have direct evidence of the configuration of the network in the wet hydrogel state. We have now reported in SI (Supplementary Figure 2) the ATR-IR spectra collected in different points of dried samples in which both typical peaks of PVA and AAc are present simultaneously (C=O stretching of AAc at 1708 cm^{-1} and OH rocking of PVA at 1325 cm^{-1}). This result, together with the transparency of the formulations and the final samples, allows concluding that the two networks can be considered homogeneously mixed.

According to the reply above, we rephrased the manuscript as follows:

L112: “The homogeneity is confirmed both by the transparency of the formulations and printed objects and by ATR-IR spectra collected in different points of dried samples, in which typical peaks of both PVA and AAc are present simultaneously (Supplementary Figure 2).”

3.6 -L141: “the high monomer reactivity caused an over-curing of the object together with the thickening of the ink”. Please explain why overcuring is a problem and, at the same time, you need a post-curing step.

We understand that the term “overcuring” is misleading. We used it to indicate an undesired curing out of the irradiated areas. For sake of clarity we avoided to use the term “overcuring” and replaced it along the text.

Considering the undesired curing out of the irradiated areas, this is a general problem that must be addressed to achieve good shape fidelity and precision. (Ref. 34) This issue, together with the high viscosity of the used resins, may lead to loss of precision. For this reason, to better control the extent of reaction, we introduced a water-soluble dye (L 101). On the other hand, in light-activated 3D printing, usually, the process is performed just to gelation to minimize the irradiation. Thus, a post-curing step after removing the unreacted resin is usually required to complete the polymerization and strengthen the printed object (Ref. 35).

We added to the manuscript the following sentences to explain better this issue:

L149: “At last, the high photoreactivity of the monomer contributed to the thickening of the ink and eventually uncontrolled polymerization, with loss of printing precision.”

L270: “The objects were then post-cured after removing the unreacted resin to complete the polymerization and to strengthen the printed objects. Post-curing was performed in a UV chamber with a medium-pressure mercury lamp (Asiga Flash or RobotFactory UV oven) for 2 minutes.”

3.7 -L162-164 and L171: I find very elegant the production of samples with different colors, as healing could be then easily seen. But the addition of the dye has an essential impact in the photopolymerization reaction (penetration of the light, resolution...). Other features of photopolymerization (especially conversion) should be affected when dye is changed. The different conversion/cross-linking density could cause that migration of dye is only observed in one direction (red dye to green area), and purple color is only seen at one side of the healed cut (Fig 3c-inset)(and so PVA chains should do). Change of dyes is only done for a better view of the healing area, but I think that authors should explain this.

As correctly pointed by the reviewer, the use of different colorants was performed only to give a visual impression of the different samples rejoined and healed, and we agree with the reviewer regarding possible competitive absorption of the dyes. However, samples printed with different dyes achieved similar conversion after post-curing as shown in the following ATR spectra normalized with respect to the peak centered at 1088 cm^{-1} .

We believe that the color impression was just a matter of diffusion of the dye in the network. In order to clarify this point, we performed an additional experiment (see pictures below, added in the SI file as Supplementary Figure 8). We printed cuboid samples (25mm x 10 mm x 5 mm) with formulations containing the two different dyes and without the dye. We cut the colored samples in half, and we attached the two different halves to the undyed sample, from where we cut the two ends to expose fresh surfaces. The samples were stored in a sealed vessel to minimize water evaporation.

As can be seen, methyl red migrates more rapidly than brilliant green in the central part. The two molecules show different diffusivity in the hydrogel, probably due to their hindrance, nature, and affinity with the matrix, so they need a different time to diffuse along the same length. After 24 hours, methyl red shows visible diffusion in the central part, while brilliant green does not. After 15 days, the diffusion gradients are in contact, with a purple color similar to samples showed in the manuscript (Supplementary Figure 8a). The prolongation of the diffusion path (Supplementary Figure 8b) helps to show the differences in the diffusion kinetics of the two dyes. It can be concluded that the diffusion of methyl red is faster than brilliant green.

To clarify this issue, we included the above explanation in the SI and added the following sentence in the revised manuscript:

L184: “The inter-diffusion ability beyond the interface plan is revealed (Fig. 3c inset) by the purple color at the interface, which results from mixing the red and green dyes (Supplementary Figure 8a). **Note that the two dyes have different diffusivity in the hydrogel, and therefore the purple color appears to be only at one side of the interface (Supplementary Figure 8b).**”

3.8 - Materials and methods: authors use three different irradiation systems (385nm led, non-filtered Hamamatsu UV lamp, and medium pressure Hg lamp). Please justify why some experimental values obtained from one are used to fix experimental conditions in other one. And why conclusions obtained with Hg irradiation systems could be extrapolated to 385nm-led irradiation one.

We agree with the reviewer that different light sources may lead to different photocuring kinetics and slightly different materials, but the different irradiation systems were used for different purposes and are not intended to be directly related. Photorheology could only be performed using an optical fiber light source, and it was used as a supplementary test to have a first indication of the kinetics of the different formulations. However, the correct printing parameters must be determined experimentally in the 3D printer. All the printed samples were fabricated by a 385 nm LED light-source printer, and all underwent the same post-curing process with a medium pressure Hg lamp. We are aware that there are differences in the cross-linking density between layered and bulk samples, with the layer-by-layer fabrication providing a better mechanical homogeneity in the structure, as indeed stated in the manuscript (L 203).

3.9 -L245: “All the printed samples were rapidly cleaned with a cloth wetted with ethanol” How these complicated shapes could be cleaned with a wetted cloth?

We are aware that manual cleaning could be problematic for very complex architectures. Therefore, the cleaning method is based on first blowing compressed air onto the sample, removing most of the residual resin. For some structures, we needed to remove some remaining residues, and this was done manually by using a cloth with ethanol. It should be noted that such cleaning processes are typical for a variety of printing technologies. Therefore, the cleaning procedure has been updated in the experimental section as follows:

L268: “After printing, the samples were cleaned using compressed air to remove the unpolymerized resin, and for some structures that have residual resin, it was manually removed with a cloth wetted with ethanol.”

REVIEWER COMMENTS

Reviewer #1 (Remarks to the Author):

This manuscript describes the formulations and mechanical testing of 3D printed polyvinylalcohol, polyacrylic acid, poly(ethylene glycol diacrylate) conetworks. Photorheology confirmed rapid crossover times and a plateau modulus below the Dahlquist criterion in the optimized formulation. The inclusion of organic dyes enabled higher resolution printing through the absorbance of the printers lightsource. A commercial digital light processing printer achieved geometrically complex parts with resolutions comparable to commercial materials. 3D printed parts displayed self-healing properties. Tensile testing revealed a 72% recovery in strength after a 12-hour healing period in humid environments. Self-healing was only observed in the formulations with the largest weight % loading of PVA. While this work displays a novel class of 3D-printable self-healing materials, it does not provide enough fundamental evidence to explain the structure property relationships that impart the self-healing behaviors of these materials. Without further fundamental investigations, this paper is not suitable for publication in Nature Communications.

Comments:

- The authors must provide and support a mechanism for the self-healing of their materials.
- The authors must quantify the diffusion across the fracture point to further investigate the mechanisms of self-healing.
- The authors must address the fact that their materials are below the Dahlquist Criterion and thus likely to behave as pressure sensitive adhesives.
- The authors must investigate their material's propensity to adhere to itself at locations other than the fracture site.
- The authors must not classify their materials as semi-IPNs or IPNs without the necessary morphological characterizations.
- The authors must probe crosslink density effects on self-healing across their different formulations.
- The authors should not propose a hydrogen bonding mechanism at this time as the covalent polyacrylic acid network is also capable of hydrogen bonding.
- The authors should investigate different PVA molecular weights and the effects of entanglements in solution on the material self-healing.
- The authors should not claim the self-healing does not require external stimuli when it is necessary for the materials to be in humid environments to self-heal.
- The authors should not suggest the material viability for energy harvesting without expanding upon the material properties that are advantageous for energy harvesting and displaying that these materials match those criteria.
- The authors should carefully proofread their manuscript for grammar and correct scientific style.

Reviewer #2 (Remarks to the Author):

I am happy that you have responded to all my queries and request for amendments to the article. I now feel that this article is acceptable for publication.

Reviewer #3 (Remarks to the Author):

The new version of the manuscript is clearly improved and I recommend its publication in its actual form.

My questions have been adequately answered/completed/corrected, and I'm satisfied with the result.

In my opinion the findings of the work are of high interest

Reviewer #1 (Remarks to the Author):

1- The authors must provide and support a mechanism for the self-healing of their materials.

A similar response was given in our previous rebuttal to the reviewer who asked about the self-healing mechanism (question 1.3 of the first reviewers' reply). In PVA, Self-healing behavior was explained both by polymer chain diffusion (Ref. 45) and hydrogen bonding across the interface (Ref. 45, 46,47), which both can be useful for our purpose.

Furthermore, the self-healing mechanism in similar systems has already been presented in the following studies cited in the manuscript.

Ref. 45 10.1021/mz300451r (2012) hydrogel based on neat PVA

Ref. 46 10.1021/acs.langmuir.5b03474 (2015) Hydrogel based on PEGDA/PVA

Ref. 47 10.1021/acsami.6b05627 (2016) Hydrogel based on PVA/Acrylamide and AAC

Ref. 48 10.1016/j.cej.2019.04.077 (2019) hydrogel based on PVA/PAAc ionically cross-linked

To explain more in detail the SH mechanism, the manuscript was revised as follows, including the addition of a new Supplementary Figure and a new Supplementary Video (Supplementary Figure 10 and Supplementary Video 2):

In the section “system design”, L102: “PVA was selected since it is a biocompatible and well-known material to produce efficient SH hydrogels due to chains interdiffusion and the presence of hydroxyl groups that are able to form extensive hydrogen bonding, as already described in several publications^{45-48,50,51}.”

In the section “Self-healing of 3D printed samples and mechanical characterization” L197: “The self-healing can be attributed to the efficient adhesion at the rejoined surface, which is due to the formation of hydrogen bonds across the interface both by carboxylic and mostly hydroxyl groups.^{45,54} Furthermore, additional experiments showed that the SH on freshly cut surfaces is indeed more efficient than simple adhesion forces (Supplementary Video 2). This behavior can be explained since a considerable amount of hydrogen bond-forming groups available on the freshly cut surface is expected, strengthening the interactions at the interface.”

L209: “Tensile tests were performed at different restoring times to have a quantitative evaluation of the SH with time (Fig 4.a). As seen, the system yields a considerable recovery already within the first two hours, and the recovery slightly increases with time, reaching a plateau in the achievable restoration after 12 hours of contact (fig 4.b). The increase with time may be a result of the PVA interdiffusion.⁴⁵ The possibility of diffusion beyond the interface plan was also evaluated qualitatively using two water-soluble

dyes in different sides. After rejoining, purple color is observed at the interface (Fig. 3c, inset), which results from mixing the red and green dyes (Supplementary Figure 8a). Note that the two dyes have different diffusivity in the hydrogel, and therefore the purple color appears to be only at one side of the interface (Supplementary Figure 8b). It should be noted that the strong similarity between hydrogen bonding among PVA chains and the H-bonds in water in FT-IR (broad peak around 3200 cm^{-1}) makes impossible to quantitatively analyze the polymer chains diffusion or the formation of hydrogen bonding across the surfaces (Supplementary Figure 10). In fact, the bands of hydroxyl groups are hidden by the dominant bands of water (65 wt% water in the hydrogel), as already reported in literature⁴⁵.

Supplementary Figure 10. ATR-IR spectra comparison of an as-printed sample (wet sample) and a dried sample (dry sample). It can be noted that the bands of hydroxyl groups of the PVA in the wet sample are hidden by the dominant bands of water (broad peak around 3200 cm^{-1}).

2- The authors must quantify the diffusion across the fracture point to further investigate the mechanisms of self-healing.

In order to have a qualitative demonstration of the diffusion through the healed surface, in our previous experiments we used two different dyes (Supplementary Figure 9). The diffusion can be ascribed to dye mobility within the network, so quantifying this diffusion (for instance, by means of UV-Vis spectroscopy) would result only in a relative evaluation. It should be noted that quantifying the diffusion by following dye movement would require a chemical linking of each dye to the polymer, which would also affect the properties of the polymers. Therefore, it will not provide quantitative information on the diffusion of the PVA and PAA molecules in the system.

However, as previously stated, PVA-based systems were thoroughly studied in the literature, and the SH mechanism is commonly accepted as resulting from both interdiffusion and hydrogen bonding (Ref. 45). To provide a better explanation of the healing mechanism involved, we modified the manuscript as described in reply to comment 1.

3- The authors must address the fact that their materials are below the Dahlquist Criterion and thus likely to behave as pressure-sensitive adhesives.

The Dahlquist Criterion is a practical rule-of-thumb for the flowability of pressure-sensitive adhesives (PSA) to ensure good adhesion with all the features of two different rough surfaces. The Dahlquist Criterion states that elastic shear modulus G' of a PSA should be $<0.1\text{MPa}$ (Parente, M. E., Ochoa Andrade, A., Ares, G., Russo, F. & Jiménez-Kairuz, Á. Bioadhesive hydrogels for cosmetic applications. *Int. J. Cosmet. Sci.* 37, 511–518 (2015), now added as Ref. 57). Our material, indeed, shows rheological properties below the Dahlquist criterion, therefore the mechanism of self-healing can also be influenced by surface self-adhesiveness at the interface. We would like to point out that this aspect does not exclude that the material is self-healing, since macroscopic self-repairing can be defined as the recovery of the initial mechanical properties (Ref. 36,40).

To clarify this point, we added the following paragraph:

L202: “It can be argued that the self-healing mechanism is similar to the behavior of pressure-sensitive adhesives (PSA) between two separate surfaces. A common measure for this effect is the Dahlquist criterion, which requires that the elastic shear modulus G' should be $< 0.1\text{MPa}$ ⁵⁷. Our material shows rheological properties below the Dahlquist criterion, and therefore the mechanism of self-healing can also be influenced by surface self-adhesiveness at the interface. However, this aspect does not exclude that the material is self-healing, since macroscopic self-repairing can be defined as the recovery of the initial mechanical properties^{36,40}. Indeed, the restoration of complex architectures with overhanging features was successfully achieved (Fig. 3.c).”

4- The authors must investigate their material’s propensity to adhere to itself at locations other than the fracture site.

In order to evaluate this point, we performed additional experiments included in the revised manuscript as supporting video file (Supporting Video 2).

From the video, it is evident that adhesion of cut surfaces is more efficient than on other parts (for instance, sustain gentle pulling).

The following paragraph addressing this issue was added to the revised manuscript:

L199: “Furthermore, additional experiments showed that the SH on freshly cut surfaces is indeed more efficient than simple adhesion forces (Supplementary Video 2). This behavior can be explained since a

considerable amount of hydrogen bond-forming groups available on the freshly cut surface is expected, strengthening the interactions at the interface.”

5- The authors must not classify their materials as semi-IPNs or IPNs without the necessary morphological characterizations.

This issue was already discussed in response to questions 2.5 and 3.5 of the first reviewers' comments, as follows:

Combining two different polymers that are not covalently bound together in one system was reported and recognized as interpenetrated networks (Dragan, E. S. Design and applications of interpenetrating polymer network hydrogels: A review, Chem. Eng. J. 243, 572–590 (2014), now added as Ref. 24). Interpenetrating network hydrogels (IPNs) are composed of separate networks physically entangled within each other without any covalent bonds between them, with at least one of them being cross-linked within the immediate presence of the other. According to the definition reported in the reference mentioned above, our system is a sequential semi-IPN since the precursors of the chemical covalent network are added to a linear polymer solution that is entrapped in the cross-linked matrix after polymerization. We called the system “double network” to highlight the different nature of the constituents, but in compliance with the specific definition, it was not completely consistent. Therefore, we changed the term “double network” throughout the manuscript to semi-IPN.

In the previous revision, the text was modified accordingly:

L49: “A convenient strategy to gather 3D shaping (printability) and self-healing properties consists in designing hydrogels with an interpenetrated network (IPN)^{21,22}. Those networks combine a rigid and robust frame (chemical bonds, generally non-reversible) with a much weaker one (physical bonds, mostly reversible)²³, displaying new tailored features, such as improved toughness and flexibility while retaining the characteristics of both constituents simultaneously²⁴. It has been shown that IPN hydrogels match extrusion-based 3D printing material requirements^{5,25,26}.”

L97: (Results-System design) “The system was designed as a sequential semi-IPN since the precursors of the chemical covalent network are added to a solution containing a linear polymer that is entrapped in the cross-linked matrix after polymerization²⁴.”

Regarding the tests, it is impossible to have direct evidence of the configuration of the network in the wet hydrogel state since morphological characterizations would not be useful to distinguish chains of different nature but similar in their backbone. We have now reported in SI (Supplementary Figure 3) the ATR-IR spectra collected in different points of dried samples in which both typical peaks of PVA and AAc are present simultaneously (C=O stretching of AAc at 1708 cm⁻¹ and OH rocking of PVA at 1325 cm⁻¹). This result, together with the transparency of the formulations and the final samples, allows concluding that there are no separate phases, and so the two networks can be considered homogeneously mixed.

The following sentence was added in the previous revision:

L118: “The homogeneity is confirmed both by the transparency of the formulations and printed objects and by ATR-IR spectra collected in different points of dried samples, in which typical peaks of both PVA and AAc are present simultaneously (Supplementary Figure 3).”

6- The authors must probe cross-link density effects on self-healing across their different formulations.

We agree with the reviewer that the cross-linking density should affect self-healing. The chemical cross-linking is obtained only using poly (ethylene glycol) diacrylate (PEGDA), which is used in a constant molar ratio of 1:1000 to acrylic acid. Throughout all the formulations, we used only one concentration of PEGDA, which was the lowest concentration of cross-linker that enabled a successful printing. A higher concentration of PEGDA (so higher cross-linking densities) still enabled printing 3D objects, but they were without SH properties.

Regarding the cross-linking density, being our material a semi-IPN polymer network, we can provide apparent values, including physical and chemical bonds. Those were calculated from the elastic shear modulus at the plateau of the formulations, using the formula from (Noè, C., Tonda-Turo, C., Chiappone, A., Sangermano, M. & Hakkarainen, M. Light Processable Starch Hydrogels. *Polymers* (Basel). 12, 1359 (2020), now Ref 56):

$$v_e = \frac{G'_p N_A}{RT}$$

Where v_e is the network cross-linking density, G'_p is the shear storage modulus in the frequency-independent plateau region, R is the universal gas constant, T is the temperature, and N_A is the Avogadro's number.

The apparent cross-linking densities for each formulation were:

Formulation	Apparent cross-linking density (m ⁻³)
PVA 0	4.6 x 10 ²³
PVA 0.2	1.0 x 10 ²⁴
PVA 0.4	1.5 x 10 ²⁴
PVA 0.6	2.6 x 10 ²⁴
PVA 0.8	4.0 x 10 ²⁴

In our case, PVA is the active mending agent in the hydrogel, so the higher it is, the concentration better are the SH properties, regardless of the apparent entanglement/cross-linking density.

Therefore, the manuscript was revised accordingly, and the following text and a supplementary table were added:

L146: “In the printing compositions, we used only one concentration of the chemical cross-linker, PEGDA, at a molar ratio of 1:1000 to acrylic acid, which was the lowest concentration of cross-linker that enabled a successful printing. A higher concentration of PEGDA (and then higher cross-linking density) still enabled printing 3D objects, but affected the SH properties. The apparent values for cross-linking, including physical and chemical bonds, were estimated from the elastic shear modulus at the plateau of the formulations, using Equation (1)⁵⁶:

$$\nu_e = \frac{G'_p N_A}{RT}$$

Where ν_e is the network cross-linking density, G'_p is the shear storage modulus in the frequency-independent plateau region, R is the universal gas constant, T is the temperature, and N_A is the Avogadro's number. It was found that for formulations without PVA, the density was 4.6 x 10²³ m⁻³, and for the highest PVA concentration, it was 4.0 x 10²⁴ m⁻³. Supplementary Table 3 shows the calculated values for the various PVA ratios.”

7- The authors should not propose a hydrogen bonding mechanism at this time as the covalent polyacrylic acid network is also capable of hydrogen bonding.

Indeed, as stated by the reviewer, the hydrogen bonds can result from both the PVA and PAA. To emphasize this point, we added the following sentence to the revised manuscript:

L108: “The selected water-soluble acrylates, AAc and PEGDA, have been widely utilized for vat photopolymerization to fabricate high-resolution objects because they provide rapid photoinduced radical photopolymerization reaction even in large amounts of water^{52,53}. Additionally, the carboxylic groups of the acrylic acid can form multiple hydrogen bonds with the PVA chains and, hence, contribute to the self-healing ability⁵⁴.”

L197: “The self-healing can be attributed to the efficient adhesion at the rejoined surface, which is due to the possible formation of hydrogen bonds across the interface both by carboxylic and mostly hydroxyl groups^{45,54}.”

8- The authors should investigate different PVA molecular weights and the effects of entanglements in solution on the material self-healing.

Indeed, we performed many preliminary studies to determine the most suitable MW and concentration of the PVA to match DLP printability and efficient self-healing. For the sake of clarity, we did not include all this data in the submitted manuscript but added it to the supplementary section.

The revised manuscript addresses this point, along with the Supplementary Figure:

L105: “Preliminary experiments were performed to evaluate the optimal MW of the PVA (Supplementary Figure 2), after which the MW of 89-98KD was selected. Lower MW did not show adequate self-healing even at a high weight percentage, 30% in water, the highest achievable that allowed suitable viscosity for DLP. Likewise, higher MW did not show acceptable efficiency, resulting from the excessive viscosity even at low concentrations”.

Supplementary Figure 2: Comparison of the healing efficiencies estimated from tensile strength at break (SH σ) and elongation at break (SH ϵ) for formulations containing PVA with different molecular weights (MW) at the highest concentration in water suitable for printing.

9- The authors should not claim the self-healing does not require external stimuli when it is necessary for the materials to be in humid environments to self-heal.

This issue is discussed in response to question 1.2 of the first reviewers' comments, as follows:

We agree with the reviewer that there are examples of water-triggered self-healing in dry systems, but in these systems, the water is added locally at the cut interfaces (see, for example, Ref. 21). In our work, the printed objects are composed of a hydrogel, in which water is a necessary and major component of the structure, and therefore, in our view, water cannot be considered as an external stimulus, as also described in the review on self-healing hydrogels (Ref. 36). Furthermore, the designed complex-shaped hydrogel showed effective adhesion immediately after rejoining (L190) without any external assistance or stimulus, except for the manual contact of the interfaces. However, the completion of the self-healing process was achieved only over time, about 12 hours (L209).

According to the reviewer's suggestion, we performed the suggested experiment, keeping the samples in a sealed container without the addition of humidity. Dumbbell-shaped specimens were 3D printed and tested in tensile tests. Self-healing properties were evaluated by testing the samples every 24 hours for five consecutive days, rejoining the samples after rupture, and keeping them in a sealed environment. The results are reported in the left figure below, which has also been added to the SI file as Supplementary Figure 11b. As can be seen, the restoration took place every time with lower efficiency, while the first healing process was similar to that obtained for the samples stored under humidity. This behavior can be explained considering the minor loss of weight that we measured during the time, as reported in the figure below on the right, also added in the SI file as Supplementary Figure 11c. This weight loss, unfortunately, could not be avoided because of water evaporation during handling and testing. These results confirm the need to keep a constant amount of water in the hydrogel to preserve the SH properties over time.

Supplementary Figure 11: left) Tensile strength and recovery percentage of samples after five separation-healing cycles, with samples stored in a sealed vessel with no controlled humidity for 24 hours while healing for every cycle. right) Comparison of weight loss during several days between samples only stored in closed and sealed vials and samples subjected to manipulation for tensile testing.

In order to clarify this point, we modified the text as follows:

L122: "Furthermore, as will be shown, the self-repair occurs without external stimuli or adhesives and is only due to internal forces because the printed material itself is already rich in functional groups able to

form hydrogen bonding and contains a large fraction of water, providing to PVA high mobility to migrate across the ruptured interface^{14,45}.”

While the manuscript was previously revised as follows:

L235: “The samples were kept in a sealed closed vessel with a humid environment to reduce water evaporation and maintain controlled conditions after-printing during the entire healing process.”

L238: “A second set of experiments was performed keeping the samples in a sealed container without controlled humidity, evidencing a lower recovery efficiency every cycle (Supplementary Figure 10b). This behavior can be explained by the loss of water during setting up the sample to measure the self-healing (Supplementary Figure 10c). As expected, keeping a constant water concentration in the hydrogel is crucial to preserve the SH properties.”

10- The authors should not suggest the material viability for energy harvesting without expanding upon the material properties that are advantageous for energy harvesting and displaying that these materials match those criteria.

Our suggestion was to highlight potentially interesting fields of application, while tailoring the properties for every use is needed according to the specific requirements. The self-healing ability in energy storage and harvesting devices is getting significant attention, as shown by the review by Chen *et al.* (reference 63).

To make the statement more general, besides the conclusion part, we rewrote the relevant sentence in the introduction as follows:

L92: “This work offers a general and easily adaptable approach to develop self-repairing hydrogels with complex three-dimensional architecture via vat photopolymerization for applications in diverse fields, ranging from biomedicine and wearable sensors to robotics and energy harvesting.”

11- The authors should carefully proofread their manuscript for grammar and correct scientific style.

We performed additional thorough proofreading of the manuscript to correct the grammar and scientific style.

Reviewer #2 (Remarks to the Author):

I am happy that you have responded to all my queries and request for amendments to the article. I now feel that this article is acceptable for publication.

Thank you so much for your very nice comment.

Reviewer #3 (Remarks to the Author):

The new version of the manuscript is clearly improved and I recommend its publication in its actual form.

My questions have been adequately answered/completed/corrected, and I'm satisfied with the result.

In my opinion the findings of the work are of high interest

We would like to thank the reviewer for the positive evaluation of our work.

REVIEWERS' COMMENTS

Reviewer #1 (Remarks to the Author):

The authors did a good job addressing previous comments and the manuscript is suitable for publication in Nature Communications

Reviewer #1 (Remarks to the Author):

The authors did a good job addressing previous comments and the manuscript is suitable for publication in Nature Communications

Reply: On behalf of all authors, I would like to thank the reviewer for the positive evaluation of our work.

Sincerely

Ignazio Roppolo